# CATaDa reveals global remodelling of chromatin accessibility during stem cell differentiation in vivo

Gabriel N Aughey, Alicia Estacio Gomez, Jamie Thomson, Hang Yin, Tony D Southall*

Department of Life Sciences, Imperial College London, London, United Kingdom

**Abstract** During development eukaryotic gene expression is coordinated by dynamic changes in chromatin structure. Measurements of accessible chromatin are used extensively to identify genomic regulatory elements. Whilst chromatin landscapes of pluripotent stem cells are well characterised, chromatin accessibility changes in the development of somatic lineages are not well defined. Here we show that cell-specific chromatin accessibility data can be produced via ectopic expression of *E. coli* Dam methylase in vivo, without the requirement for cell-sorting (CATaDa). We have profiled chromatin accessibility in individual cell-types of Drosophila neural and midgut lineages. Functional cell-type-specific enhancers were identified, as well as novel motifs enriched at different stages of development. Finally, we show global changes in the accessibility of chromatin between stem-cells and their differentiated progeny. Our results demonstrate the dynamic nature of chromatin accessibility in somatic tissues during stem cell differentiation and provide a novel approach to understanding gene regulatory mechanisms underlying development.

DOI: https://doi.org/10.7554/eLife.32341.001

## Introduction

During the development of a multicellular organism, gene expression is tightly regulated in response to spatially and temporally restricted signals. Changes to gene expression are accompanied by concomitant changes to chromatin structure and composition. Therefore chromatin states vary widely across developmental stages and cell types. Functional regions of a genome, including promoters and enhancers, can be identified by their relative lack of nucleosomes. These regions of 'open chromatin' can be assayed by their accessibility to extrinsic factors. Consequently, chromatin accessibility profiling techniques are commonly used to investigate chromatin states (reviewed in [*Tsompana and Buck, 2014*]). Chromatin is highly accessible in pluripotent cell types such as embryonic stem (ES) cells, but is compacted following differentiation (*Meshorer and Misteli, 2006*). It has been suggested that this open chromatin represents a permissive state to which multiple programmes of gene regulation may be rapidly applied upon differentiation (*Gaspar-Maia et al., 2011*).

The nature of chromatin accessibility across different developmental stages in vivo is less well understood. Imaging studies have been used to demonstrate gross changes to chromatin structure, for example changes to the distribution of heterochromatin have been observed in post-mitotic cells (*Francastel et al., 2000*; *Le Gros et al., 2016*). Molecular studies investigating chromatin states in vivo during development have tended to utilise heterogeneous tissues due to the fact that profiling the epigenome of individual cell types frequently requires physical isolation of cells or nuclei, which can be laborious and prone to human error (*McClure and Southall, 2015*). Therefore, there is a lack of information regarding cell-type-specific changes to chromatin states in *in vivo* models. Whilst recently developed methods such as ATAC-seq have become popular and address many of the limitations inherent to earlier techniques such as DNAse-seq (i.e. requires fewer cells and increased

*For correspondence:
t.southall@imperial.ac.uk

Competing interests: The authors declare that no competing interests exist.

**eLife digest** For an embryo to successfully develop into an adult animal, specific genes must act in different types of cells. Though all the cells have the same genes encoded within their DNA, looking at the way that the DNA is packaged can indicate which parts of the DNA are important for that particular cell type. If regions of DNA are "open" one can infer that those regions are actively involved in gene regulation, whereas "closed" regions are considered less important.

It is currently difficult to determine which parts of the DNA are open within an individual cell type in a complex organ, such as the brain. Existing methods require the cells to be physically isolated from the tissue, which is technically challenging.

To overcome this issue, Aughey et al. have now developed a method that does not require isolation of the cells. The new technique involves using genetic engineering to introduce an enzyme called Dam into specific cell types in living fruit flies. This enzyme adds a chemical label on regions of open DNA, which can then be detected. Aughey et al. tested this technique on various cells of the developing brain and gut, and were able to see differences in the openness of DNA that corresponded to the action of genes that are important in each cell type. The data also contain trends that help to understand the role of open DNA in development. For example, mature cells were shown to overall have less open DNA than the stem cells that divide to generate them.

Aughey et al. hope their new technique will be of use to other researchers working with either fruit flies or mammalian tissues. The knowledge that scientists will gain from identifying how open DNA contributes to gene regulation, in both healthy and diseased tissues, will further our understanding of human development and the biology of diseases such as cancer.

DOI: https://doi.org/10.7554/eLife.32341.002

assay speed), these techniques still require the physical separation of cells and isolation of genomic DNA before chromatin accessibility is assayed (*Buenrostro et al., 2013*).

It has been suggested that ectopic expression of untethered DNA adenine methyltransferase (Dam) results in specific methylation of open chromatin regions whilst nucleosome bound DNA is protected (*Wines et al., 1996*; *Bulanenkova et al., 2007*; *Boivin and Dura, 1998*; *Singh and Klar, 1992*). However, the efficacy of using Dam methylation for chromatin accessibility profiling on a genomic scale is not clear. Furthermore, expression of Dam in a cell-type-specific manner, at levels low enough to avoid toxicity and oversaturated signal, has not been possible until now.

Transgenic expression of fusions of Dam to DNA-binding proteins is a well-established method used to assess transcription factor occupancy (DNA adenine methyltransferase identification - DamID) (*van Steensel and Henikoff, 2000*). Recently, it was demonstrated that DamID could be adapted to profile DNA-protein interactions in a cell-type-specific manner by utilising ribosome re-initiation to attenuate transgene expression (*Marshall et al., 2016*; *Aughey and Southall, 2016*; *Southall et al., 2013*). This technique is referred to as Targeted DamID (TaDa). Here, we take advantage of TaDa to express untethered Dam in specific cell types to produce chromatin accessibility profiles in vivo, without the requirement for cell separation. We show that **C**hromatin **A**ccessibility profiling using **Ta**rgeted **Da**mID (CATaDa) yields comparable results to both FAIRE and ATAC-seq methods, indicating that it is a reliable and reproducible method for investigating chromatin states. By assaying multiple cell types within a tissue, we show that chromatin accessibility is dynamic throughout the development of *Drosophila* central nervous system (CNS) and midgut lineages. These data have also enabled us to identify enriched motifs from regulatory elements that dynamically change their accessibility during differentiation, as well as to identify functional cell-type-specific enhancers. Finally, we show that compared to their differentiated progeny, somatic stem cell Dam-methylation signals are more widely distributed across the genome, indicating a greater level of global chromatin accessibility.

## Results

### CATaDa produces chromatin accessibility profiles comparable to that of ATAC and FAIRE-seq in *Drosophila* eye discs

We reasoned that low-level expression of transgenic *E. coli* Dam, using tissue-specific GAL4 drivers in *Drosophila*, would specifically methylate regions of accessible chromatin exclusively in a cell-type of interest. Detection of these methylated sequences could yield chromatin accessibility profiles for defined cell populations in vivo (*Figure 1*). To determine if CATaDa produces an accurate reflection of chromatin accessibility, we compared data acquired using this approach with commonly used alternative techniques. A recent study generated ATAC and FAIRE-seq data from *Drosophila* imaginal eye discs (*Davie et al., 2015*). Using CATaDa, we expressed *E. coli* Dam in the eye disc of *Drosophila* third instar larvae so that we could compare *Dam* methylation profiles to these previously collected data.

Chromatin accessibility profiles produced with CATaDa in the eye disc were highly reproducible between replicates ($r^2 = 0.947$) (*Figure 2—figure supplement 1*). CATaDa profiles showed good agreement with data produced with ATAC-seq and FAIRE-seq. Visual inspection of the data showed that many regions of accessible chromatin identified by ATAC and FAIRE are also represented by CATaDa, whilst condensed regions are reliably inaccessible (*Figure 2A,B*). We also observe that CATaDa profiles exhibited features consistent with chromatin accessibility. For example, open chromatin is enriched at transcriptional start sites (TSS) (*Figure 2C*).

We observe that CATaDa signal frequency increases dramatically towards the centre of ATAC or FAIRE peaks (*Figure 2D*).The overlap of Dam identified peaks with ATAC and FAIRE peaks is 48.6% and 49.4%, respectively (In comparison, 55.9% of ATAC peaks are also identified in FAIRE data – *Figure 2E*). A Monte Carlo simulation determined that this is a highly significant overlap ($p < 1 \times 10^{-5}$) and peak heights at shared ATAC and CATaDa peaks show significant correlation ($p < 1 \times 10^{-16}$, $r^2 = 0.138$) (*Figure 2—figure supplement 2A*). We found that increasing the stringency of our peak calling notably decreased the number of peaks identified that coincided with ATAC peaks, but had relatively little impact on unique CATaDa peak discovery (*Figure 2—figure supplement 2B*). Given these data, we suggest that the majority of these peaks are not false positives, but are genuinely accessible sites that are not detected by ATAC-seq. Further examination of these unique peaks indicates that they are significantly smaller than the shared peaks (*Figure 2—figure supplement 2C*). We also observe that for peaks identified in either ATAC or FAIRE data that are not present in CATaDa, there is a relative lack of GATC motifs, which suggests that there may be cases in which false negatives are observed due to the limitations of the resolution achievable by Dam methylation (*Figure 2—figure supplement 3A–B*). To further investigate the differences between CATaDa and ATAC or FAIRE-seq, we investigated the detection of peaks at different genomic features. We found that whilst CATaDa identified fewer peaks than FAIRE in regions proximal to gene promoters (when compared with ATAC), CATaDa was notably better at identification of non-promoter adjacent accessible sites in ATAC data compared to FAIRE-seq (*Figure 2E*). The lack of promoter peaks identified can again be explained by the relative depletion of GATC sites upstream of TSS (*Figure 2—figure supplement 3C*).

It was previously shown that ATAC-seq and FAIRE-seq data demonstrated high chromatin accessibility at experimentally validated eye-antennal enhancers (*Davie et al., 2015*). CATaDa profiles similarly showed increased open chromatin at these regions (*Figure 3A,B*). We found that for 57.9% of FlyLight eye enhancers, a corresponding peak was called in CATaDa profiles (333 of 575 enhancers). CATaDa was comparable to FAIRE-seq and ATAC-seq which identified 48% and 68.7% respectively, of validated FlyLight enhancers as peaks (*Figure 3C*).

### CATaDa profiling shows dynamic changes in chromatin accessibility during differentiation of the nervous system

In *Drosophila*, neurons are derived from asymmetrically dividing neural stem cells (NSCs). NSC divisions produce one self-renewing daughter NSC and a ganglion mother cell (GMC), which divides once more to produce neurons or glia (*Homem and Knoblich, 2012*). To further test the technique and investigate how local and global chromatin accessibility changes during the process of nervous system differentiation, we expressed Dam in specific cells with GAL4 drivers that cover four different

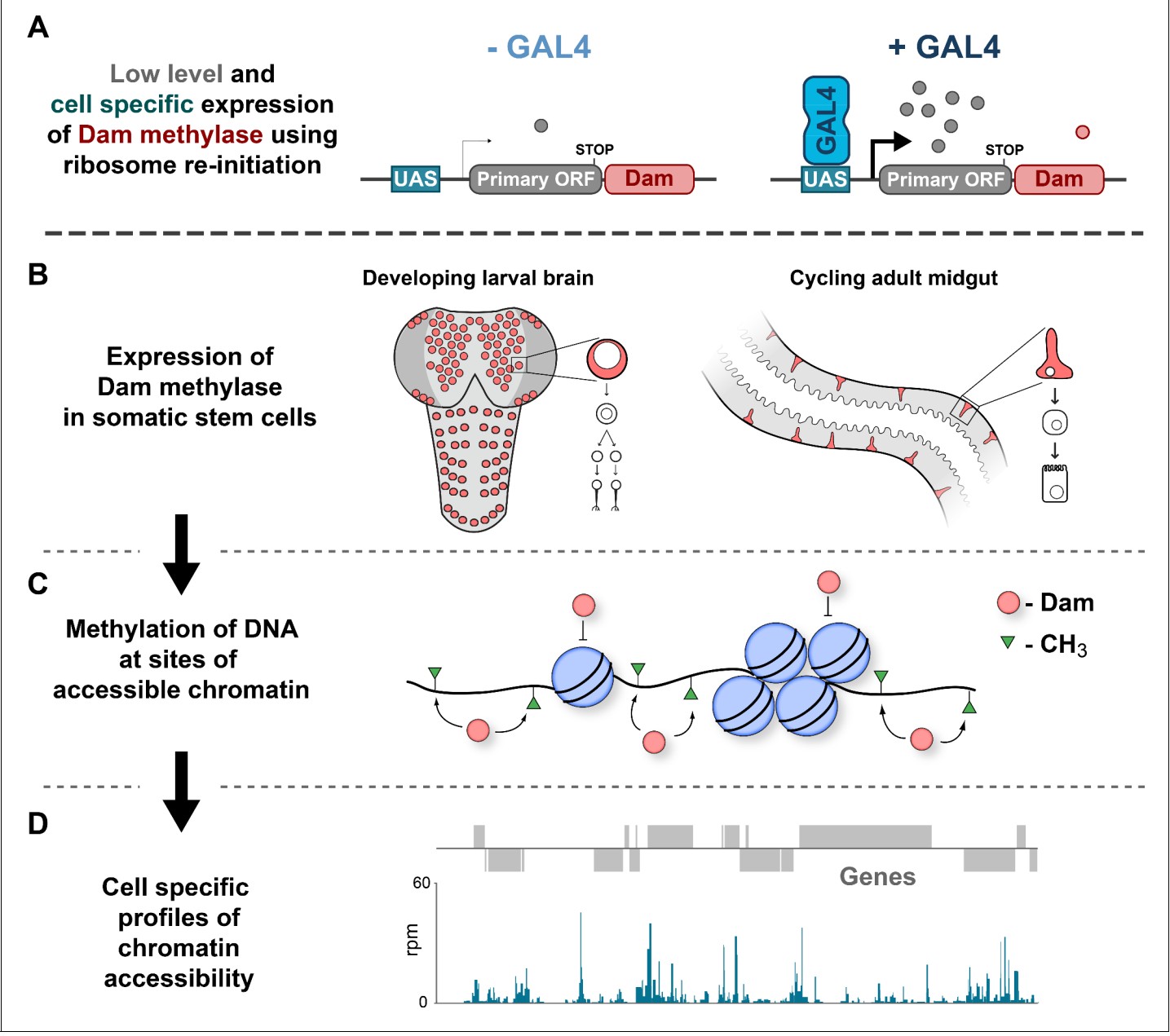

**Figure 1.** Schematic illustrating CATaDa technique. (A–B) *E. coli* Dam is expressed specifically in cell-types of interest using TaDa technique. (C) GATC motifs in regions of accessible chromatin are methylated by Dam, whilst areas of condensed chromatin prevent access to Dam thereby precluding methylation. (D) Methylated DNA is detected to produce chromatin accessibility profiles for individual cell-types of interest from a mixed population of cells.

DOI: https://doi.org/10.7554/eLife.32341.003

developmental stages within the lineage. These include NSCs (*worniu- GAL4*), GMCs and newly born neurons (*R71C09-GAL4* [*Figure 4—figure supplement 1B*, *Li et al., 2014*]), differentiated larval neurons (*nSyb-GAL4*), and also mature adult neurons (*nSyb-GAL4*) (*Figure 4A*).

By examining candidate genes differentially expressed during neural development, we observed that chromatin accessibility relates to gene expression in an expected manner. For example, intronic open chromatin peaks can be seen at the *bruchpilot* (*brp*) locus, in both third instar (L3) and adult neurons, whilst these peaks are reduced or absent in the progenitor cell types (*Figure 4B*). This corresponds with the expression of *brp*, which is specifically transcribed in neurons and has an

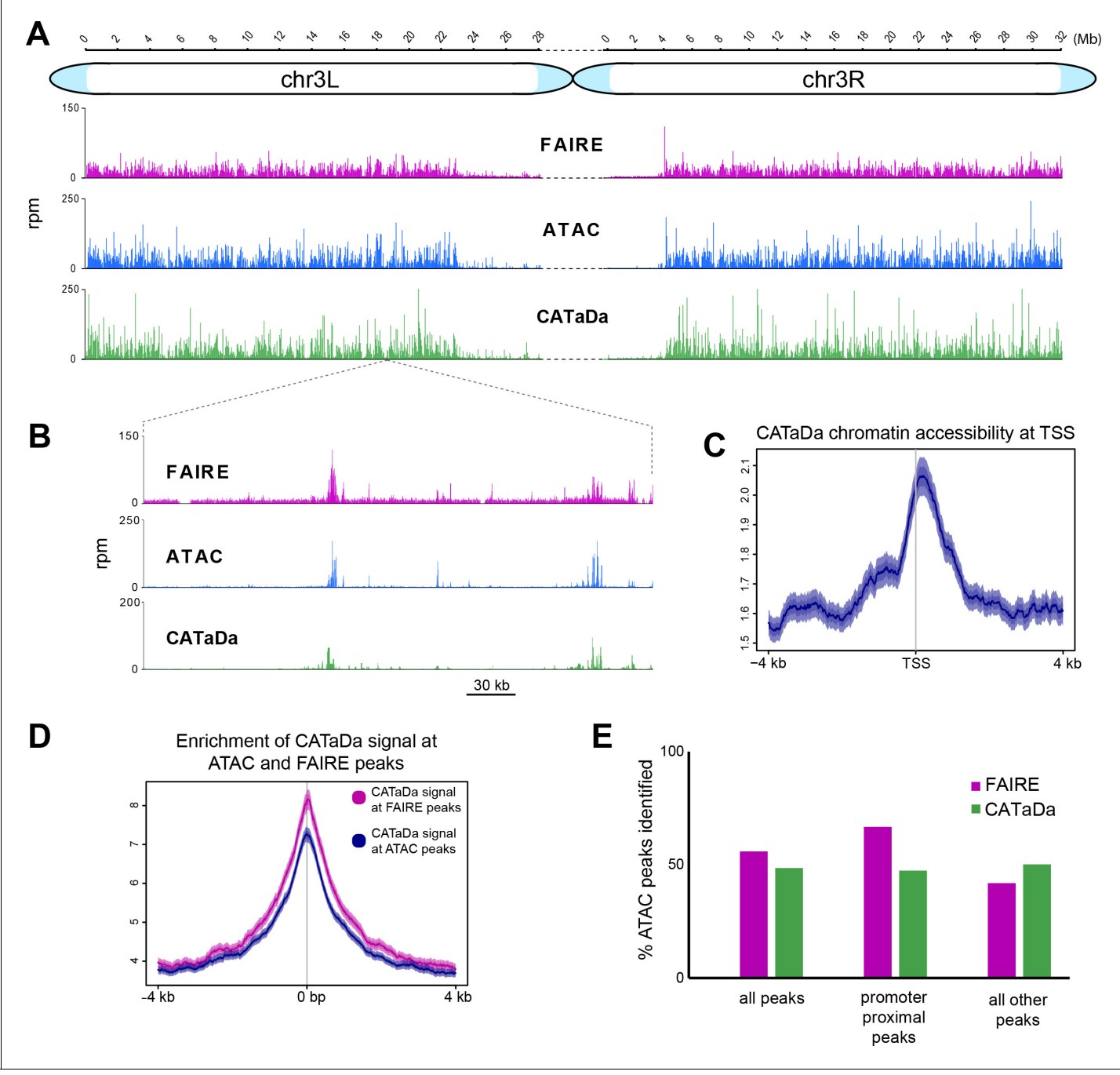

**Figure 2.** Validation of Dam chromatin accessibility profiling compared to ATAC and FAIRE-seq. (A) Chromatin accessibility across chromosome three as determined by ATAC-seq, FAIRE-seq, and CATaDa. Note the reduced amount open chromatin proximal to the centromere regions in all three datasets. y-axes = reads per million (rpm). (B) Example locus showing data obtained by FAIRE, ATAC, and CATaDa. Peaks are broadly reproducible across techniques. (C) Aggregation plot of CATaDa signal at TSS with 2 kb regions up and downstream. Aggregated signal at TSS shows expected enrichment of Dam. (D) Aggregation plot of CATaDa signal at ATAC or FAIRE peaks, indicating enrichment of CATaDa signal at these loci. (E) Identification of ATAC peaks in CATaDa or FAIRE data. CATaDa and FAIRE identify 48.6% and 55.9% of ATAC peaks, respectively. FAIRE-seq peaks overlap more frequently with promoter proximal peaks (2 kb from TSS), whilst CATaDa peaks overlap with more ATAC peaks outside of promoter regions.

DOI: https://doi.org/10.7554/eLife.32341.004

The following figure supplements are available for figure 2:

**Figure supplement 1.** Correlation between CATaDa replicates.

DOI: https://doi.org/10.7554/eLife.32341.005

*Figure 2 continued on next page*

*Figure 2 continued*

**Figure supplement 2.** Further comparison of CATaDa and ATAC data.
DOI: https://doi.org/10.7554/eLife.32341.006
**Figure supplement 3.** Frequency of GATC sites at various genomic features.
DOI: https://doi.org/10.7554/eLife.32341.007

important role in synapse function (*Wagh et al., 2006*). In contrast, the adjacent gene to *brp*, *Wnt2*, displays peaks which are most apparent in the NSC and intermediate cell types. Wnt signalling is known to be important for the control of stem cell populations, therefore, these results are also expected (*Ring et al., 2014*).

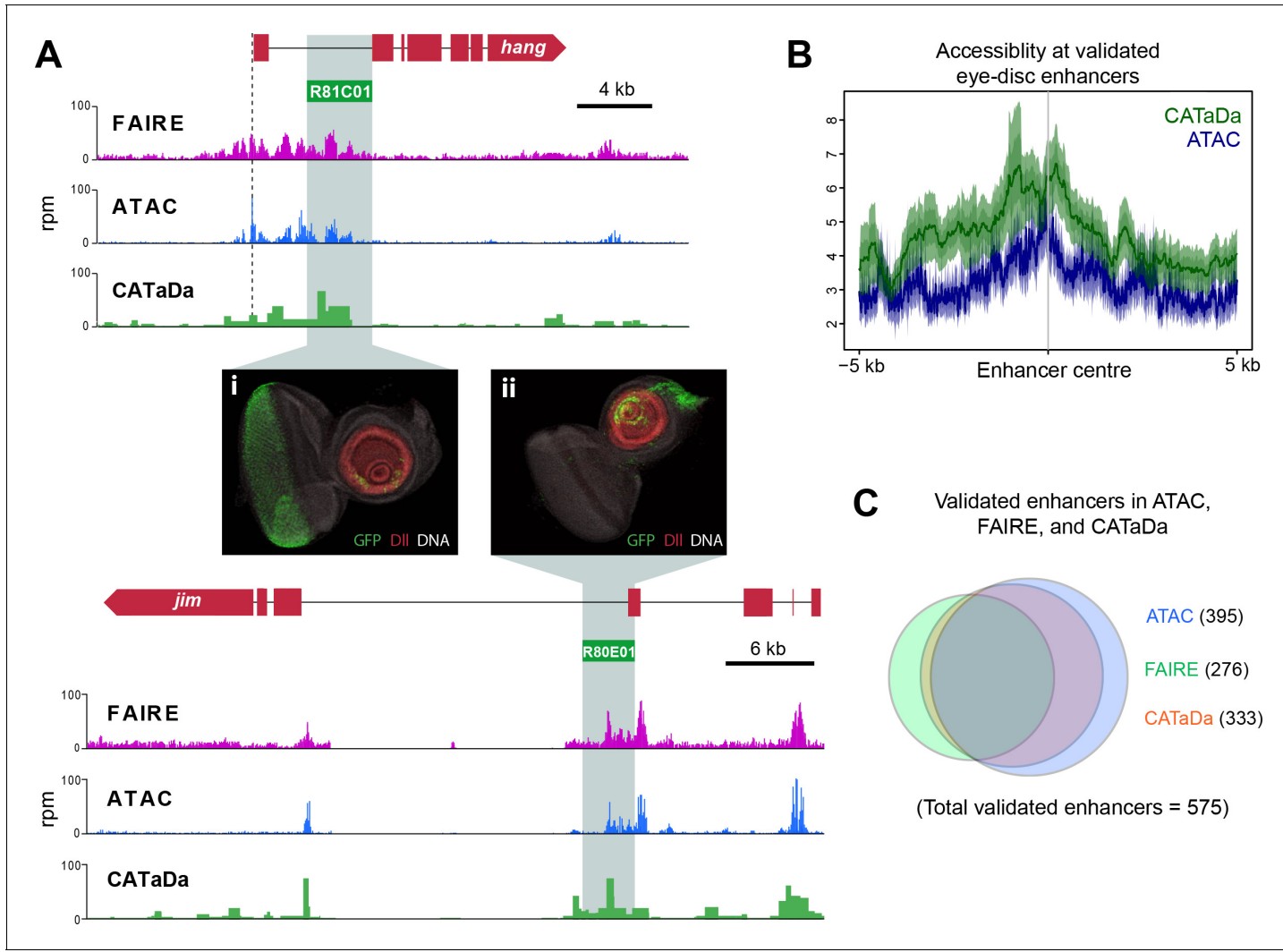

**Figure 3.** Identification of validated imaginal disc enhancers with CATaDa. (A) Example loci showing data obtained by FAIRE, ATAC, and CATaDa. Peaks are broadly reproducible across techniques. Flylight enhancers with validated expression in eye imaginal discs coincide with peaks in all three datasets. Corresponding expression pattern is shown in (i) and (ii) (eye disc images obtained from the FlyLight database [http://flweb.janelia.org/cgi-bin/flew.cgi]). (B) Aggregation plot showing average signal of ATAC (blue) and Dam (green) at 575 FlyLight enhancers with validated eye imaginal disc expression. Both techniques show increased open chromatin at these regions. (C) Venn diagram of FlyLight enhancers identified in Dam accessibility profiling, ATAC, or FAIRE-seq. The majority of enhancers identified by either ATAC or FAIRE are also found in the Dam data. Dam enhancers overlap most with ATAC (305 shared between ATAC and Dam of 575 total FlyLight enhancers).
DOI: https://doi.org/10.7554/eLife.32341.008

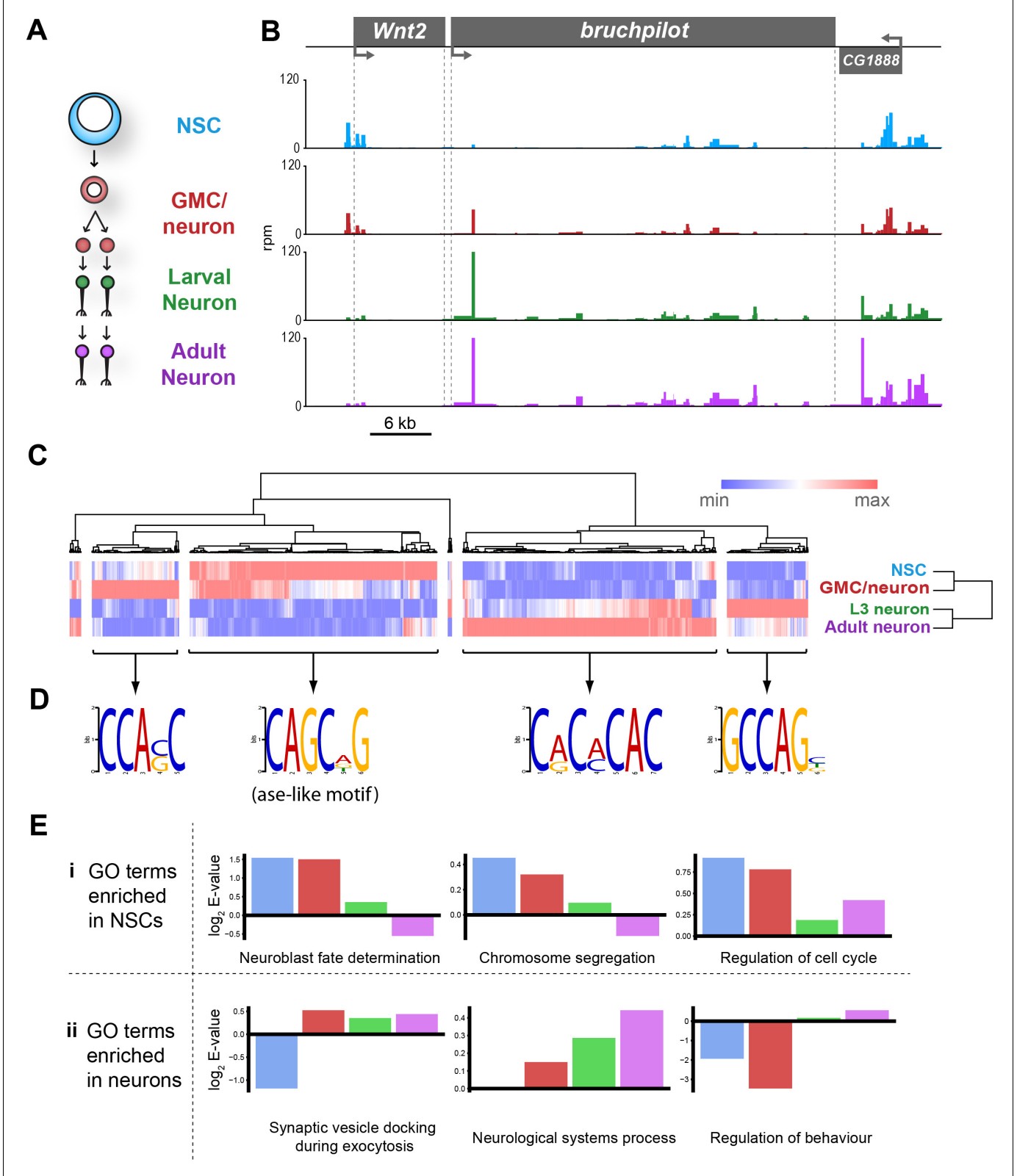

**Figure 4.** Chromatin accessibility of cell types in the CNS. (**A**) Schematic of CNS lineage progression indicating cell types examined in this study. (**B**) Example profiles resulting from Dam expression in the CNS. Genomic region encompassing *Wnt2* and *bruchpilot* genes is shown. Multiple open chromatin regions are dynamic across development. Y-axes = reads per million (rpm). (**C**) Clustering of differentially accessible regions in CNS lineages indicates two major groupings in which chromatin is most accessible in either stem cells or mature neurons. (**D**) Motif analysis using these sequences
*Figure 4 continued on next page*

*Figure 4 continued*

results in identification of expected motifs (e.g. *ase* E-box motif in stem cell accessible loci), as well as novel motifs. Most highly enriched motifs for each cluster shown. All motifs E-values < 1 × 10$^{-5}$. (**E**) log2 enrichment scores for selected GO terms in individual cell types. Clear trends can be seen as development progresses. (NSC, GMC, L3 neuron, adult neuron - from left to right). (i) GO terms are either enriched in stem cells becoming less significant as the lineage progresses or (ii) vice versa.

DOI: https://doi.org/10.7554/eLife.32341.009

The following source data and figure supplements are available for figure 4:

**Figure supplement 1.** Chromatin accessibility in neural cell types demonstrating dynamic accessibility of *R71C09* enhancer region - used to define GMC/immature neuron populations in this study.
DOI: https://doi.org/10.7554/eLife.32341.010

**Figure supplement 2.** Example loci showing dynamic chromatin accessibility in neuronal cell-types.
DOI: https://doi.org/10.7554/eLife.32341.011

**Figure supplement 3.** Top enriched motifs identified in regions of enhanced chromatin accessibility in neuronal cell types.
DOI: https://doi.org/10.7554/eLife.32341.012

**Figure supplement 4.** Further motifs enriched in cell types of the nervous system.
DOI: https://doi.org/10.7554/eLife.32341.013

**Figure supplement 5.** Top enriched GO terms for cell types of the CNS.
DOI: https://doi.org/10.7554/eLife.32341.014

**Figure supplement 5—source data 1.** Top enriched GO terms for cell types of the CNS as Excel spreadsheet.
DOI: https://doi.org/10.7554/eLife.32341.015

Similar patterns are observed at a number of other loci. At the *asense* (*ase*) locus, (a NSC-specific transcription factor), chromatin is highly accessible at the promoter and upstream intergenic region in NSCs (*Figure 4—figure supplement 2B*). This signal is considerably reduced in fully differentiated neurons in which *ase* is not expressed. Interestingly, open chromatin is still detectable in these regions in the GMCs/newly born neurons. This pattern is also observed with other NSC expressed factors such as *deadpan* (*dpn*), *CyclinE* (*CycE*) and *prospero* (*pros*) (*Figure 4—figure supplement 2*). Furthermore, GMC/newly born neuron profiles frequently show intermediate signal at these loci, indicating that functional elements required for regulation of NSC gene expression are not immediately rendered inaccessible following differentiation (*Figure 4B* and *Figure 4—figure supplement 2*).

It is to be expected that many of the functional elements marked by accessible chromatin that are important for regulating gene expression in a given neural cell type would show dynamic accessibility across the lineage (i.e. stem cell-specific enhancers would not be expected to be open in mature neurons). We examined regions of differential chromatin accessibility to determine the extent to which chromatin accessibility is changed during development of the nervous system. Hierarchical clustering of regions of chromatin with differential accessibility between cell types reveals two major clusters in which chromatin is either open in stem cells but inaccessible in neurons, or vice versa (*Figure 4C*). Intriguingly, there are other clusters where maximal chromatin accessibility is observed in either GMCs/early neurons or larval neurons. Therefore, it is not as simple as NSC accessible regions progressively closing during differentiation and neuronal regions gradually opening. There are a large number of loci that are inaccessible in NSCs, then open in the intermediate GMCs/newly born neurons stage before being rendered inaccessible again in terminally differentiated neurons (*Figure 4C*). In addition, a cluster enriched in larval neurons demonstrates that the chromatin accessibility landscape of larval neurons, although similar, is distinct from adult neurons.

Regions of open chromatin are thought to identify functional regulatory elements such as enhancers. Therefore, it is to be expected that these regions will be enriched for motifs belonging to transcription factors involved in neurogenesis. Identification of enriched motifs in sequences that were accessible in NSCs showed that expected transcription factor binding sites were highly enriched. For example, the E-box motif – CAGCNG – which is bound by the NSC proneural factor *ase* (*Figure 4D*) (*Southall and Brand, 2009*; *Jarman et al., 1993*). Regions in which open chromatin was specifically enriched in mature neurons yielded a sequence motif corresponding to the transcription factor, Ci. In all groups, sequence motifs were also identified for which no known binding partner could be identified (*Figure 4—figure supplement 3*). Analysis of further subdivision of these clusters revealed yet more novel motifs for the individual cell types examined, as well as indicating

that the ase-like motif is specifically enriched for sequences which are accessible solely in the NSCs, and not their progeny (*Figure 4—figure supplement 4*).

Gene ontology (GO) analysis of genes at which enriched chromatin accessibility was observed yielded expected biological process terms for each of the cell types examined (*Figure 4—figure supplement 5*). For example, terms such as 'neuroblast fate determination' and 'chromosome segregation' were more highly enriched in stem cells relative to neurons, whilst 'regulation of behaviour' and 'synaptic vesicle docking during exocytosis' were enriched for differentiated neurons but not NSCs (*Figure 4E*).

## Chromatin accessibility in adult midgut cell types

Having observed chromatin accessibility changes in the cells of the developing CNS, we asked whether similar patterns would be observed in adult somatic stem cell lineages. The *Drosophila* midgut contains a pool of cycling intestinal stem cells (ISCs) that persists in the adult to maintain a population of terminally differentiated cells which mediate the absorptive and secretory functions of the organ (*Jiang and Edgar, 2011*; *Nászai et al., 2015*). In contrast to neurogenesis, a single committed immature progenitor cell (enteroblast – EB) is produced from stem cell divisions, which then differentiates without further divisions to produce the mature epithelial cells of the midgut (*Ohlstein and Spradling, 2007*). To examine chromatin accessibility in the cells of the adult midgut, we expressed *Dam* in the ISCs and EBs, as well as in the terminally differentiated absorptive cells, the enterocytes (ECs)(*Figure 5A*).

As with the CNS data, we observed predictable changes in chromatin accessibility at loci for genes with variable expression in the lineage. For example, *escargot* (*esg*) a transcription factor required for ISC self-renewal (*Korzelius et al., 2014*), displays multiple peaks of accessible chromatin at the gene body and surrounding region in ISCs and EBs, whilst little signal is observed in the ECs (*Figure 5B*). In contrast the *nubbin* locus (encoding EC marker – Pdm1), displays peaks predominantly in the EC data, with relatively closed chromatin in the progenitor cell types (*Figure 5C*). As observed in the CNS, hierarchical clustering revealed two major groups in which accessible chromatin was enriched in either in the stem cells (ISCs) or differentiated cell (ECs) (*Figure 5D*). Smaller clusters were again evident in which accessible chromatin was up or downregulated exclusively in the intermediate EBs. However, this was much less pronounced than the changes observed in GMCs/early neurons of the developing CNS. This indicates that, similar to the CNS lineages, the majority of chromatin accessibility changes involved in specifying the fully differentiated cells do not occur until after EB maturation. As with the cells of the CNS, we were able to identify motifs specifically enriched in each of these groups (*Figure 5—figure supplement 1*).

ISCs and NSCs fulfil similar roles in their respective organs in the production of highly specialised functional cells. However, whilst NSCs exist for a short amount of time during fly development to produce relatively long-lived neurons that persist in the adult CNS for the animal's lifetime, the ISCs act post-developmentally to constantly replenish ECs in the adult gut. By comparing the chromatin accessibility of these two cell types, it is apparent that there are similarities in their chromatin states. At loci involved in growth or cell division, we see similar accessibility profiles across differentiation between the two tissue types (*Figure 5—figure supplement 1*). Given the similarities that we observed for individual loci between CNS and midgut lineages, we queried whether it was possible to observe trends between the cells in the two lineages on a global scale. Principal component analysis reveals two distinct clusters in which >80% of the variance is explained in the first two principal components (*Figure 5E*). These clusters represent the two distinct tissue types, (CNS and midgut) rather than immature and differentiated cells. By examining the overall correlation between all cell types we observed a number of interesting features. Firstly, as expected all cell types correlated most closely with either their direct progeny or progenitor cell (*Figure 5F*). Therefore by clustering the data we were able to recapitulate the familial relationship between the cell types of the two lineages. The greatest similarities were observed between the intermediate progenitors and their cognate stem cells ($R^2$ = 0.94/0.98 for CNS and midgut respectively). Interestingly, the greatest correlation outside of a lineage was between the two stem cell types ($R^2$ = 0.76), whilst differentiated cells exhibited only weak correlation (ISCs vs NSC, $R^2$ = 0.51). This indicates that somatic stem cell types may utilise a broadly similar chromatin landscape for the maintenance of multipotency, whilst lineage-specific variation is relatively small.

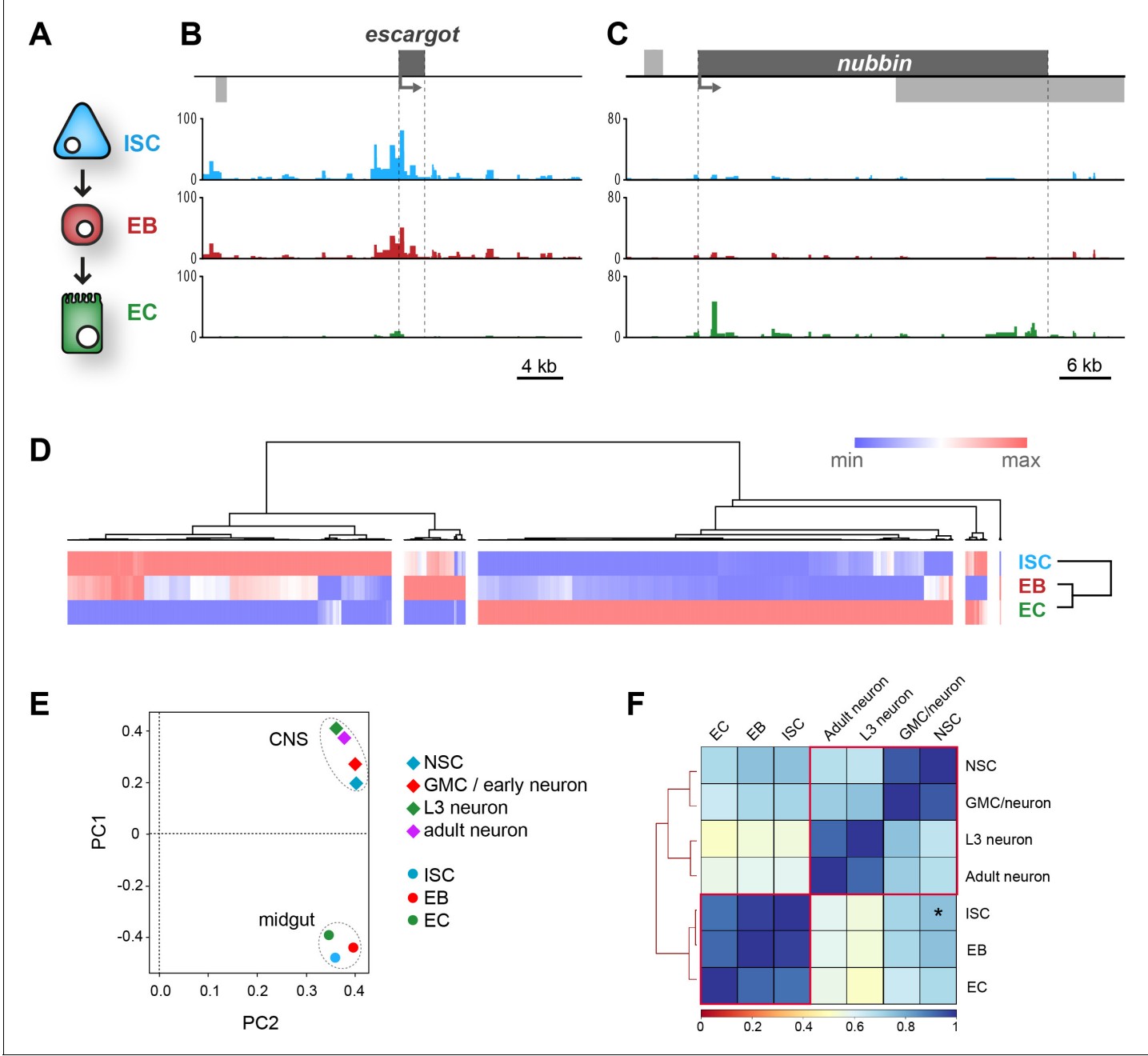

**Figure 5.** Dam chromatin accessibility profiling of cells in the adult midgut. (**A**) Schematic of midgut lineage progression indicating cell types examined in this study. (**B**) Chromatin accessibility displays expected trends at the *escargot* locus, known to be expressed exclusively in ISCs and EBs, but not ECs. Upstream promoter region shows greatest chromatin accessibility in ISCs, compared to other cell types. Similarly, dynamic peaks are observed in both 3' and 5' distal regions (putative enhancer regions), which are absent in ECs. y-axes = reads per million (rpm). (**C**) Chromatin accessibility at the *nubbin* locus, known to be expressed exclusively in ECs. y-axes = reads per million (rpm). (**D**) Hierarchical clustering of differentially accessible regions in gut cell types. Major clusters are observed in which accessible chromatin is enriched specifically in either ISCs or ECs, whilst smaller clusters indicate fewer regions with up or down-regulated accessibility in EBs. (**E**) Principal component analysis (mean of all replicates) indicates distinct groupings of both lineages. (**F**) Correlation matrix (Spearman's rank) of means of all cells in CNS and midgut lineages. Individual lineages denoted with red outline. Note relatively high correlation between NSC and ISC (Asterisk – $R^2$ = 0.76), whilst NSC correlation with EC and adult neurons are comparable.

DOI: https://doi.org/10.7554/eLife.32341.016

The following figure supplements are available for figure 5:

**Figure supplement 1.** Top enriched motifs identified in regions of enhanced chromatin accessibility in midgut cell types.

DOI: https://doi.org/10.7554/eLife.32341.017

*Figure 5 continued on next page*

*Figure 5 continued*

**Figure supplement 2.** Example loci of growth related loci with similar chromatin accessibility in CNS and midgut development.
DOI: https://doi.org/10.7554/eLife.32341.018

## Enhancer prediction from Dam accessibility data

Enhancer activity is closely linked to gene expression, therefore, many tissue-specific enhancers are required to orchestrate correct spatial and temporal transcription (*Pennacchio et al., 2013*). However, identification of functional enhancers can be challenging. Chromatin accessibility data have previously been used to identify novel enhancers (*Davie et al., 2015*; *Crawford et al., 2006*). We reasoned that it would be possible to identify genomic regions corresponding to cell-type- specific enhancers by comparing dynamically accessible regions between cell types. In support of this, we observed that the sequence covered by the *71C09-GAL4* line used in this study to profile GMCs/ newly born neurons, displayed a higher peak specifically at this region than in either the stem cell or differentiated neuron data (*Figure 4—figure supplement 1*). Interestingly, a clear peak can still be observed in the NSC data, without concomitant reporter expression. Therefore, an enrichment of accessible chromatin does not necessarily correspond to an active enhancer in a given cell type. This is consistent with previous observations that DNase hypersensitive regions are often not active enhancers (*Zhou et al., 2017*; *Thurman et al., 2012*).

We selected accessible regions with large differences between at least two cell types in the lineage, which satisfied various criteria for us to designate them as putative enhancers (see Materials and methods). We then identified available reporter lines from the Vienna tiles (VT)(*Kvon et al., 2014*) and FlyLight (*Jenett et al., 2012*) collections of GAL4 reporter lines that contained sequences encompassing our predicted enhancers upstream of a GAL4 reporter, and verified their expression in the tissues of interest. We identified enhancer-GAL4 lines in which reporter expression matched our predictions for enhancer activity. In the CNS Vienna line VT017417 and FlyLight line GMR56E07 both showed expression in the early part of the lineage in the CNS, with GFP reporter expression detectable predominantly in NSCs and GMCs (*Figure 6A,B*). This is consistent with accessible chromatin readings from our CATaDa data for these cell types in which progenitor cells displayed prominent peaks, whereas differentiated neurons did not. Similarly, we were able to detect functional cell-type-specific enhancers in the midgut. The Vienna line, VT004241, showed reporter expression predominantly in *Delta* positive ISCs (*Figure 6C*). Therefore, it is possible to use CATaDa data to identify novel cell-type-specific enhancers in multiple tissues. Overall 17 of 30 lines (57%) tested showed GFP expression in tissues and cell types that closely matched our predictions based on CATaDa data. This is in line with the rate of enhancer prediction by previous methods (*Kvon et al., 2014*).

## Distribution of sequencing reads reveals changes in global chromatin accessibility during lineage development

It is commonly accepted that the global accessibility of chromatin in a given cell type correlates broadly with potency. Whilst there are multiple lines of evidence showing that pluripotent cells have high levels of open chromatin, little data exist to support this idea in somatic multipotent stem cells and their progeny. We reasoned that by examining the distribution of normalised mapped sequencing reads (for each GATC fragment) we would be able to determine the nature of open chromatin in a given cell type.

Initial examination of read distributions indicated that differentiated neurons had fewer GATC fragments with lower read counts (*Figure 7—figure supplement 1A*). However, having a limited number of replicates prevented a thorough statistical analysis of this relationship. To increase the significance of our analysis of read distributions, we decided to incorporate further neuronal and NSC datasets into our analysis, which were available to us as control data from existing DamID studies ([*Marshall et al., 2016*] and unpublished data). Including our previously described neuronal data, we examined the distribution of twelve adult and larval neuronal Dam accessibility datasets (derived from individual post-mitotic neuronal subtypes, cholinergic, GABAergic, and glutamatergic) and compared to data derived from NSCs.

From these distributions it is apparent that majority of GATC fragments in the genome have very few corresponding mapped reads in all samples, whilst fragments having over ~10 reads per million

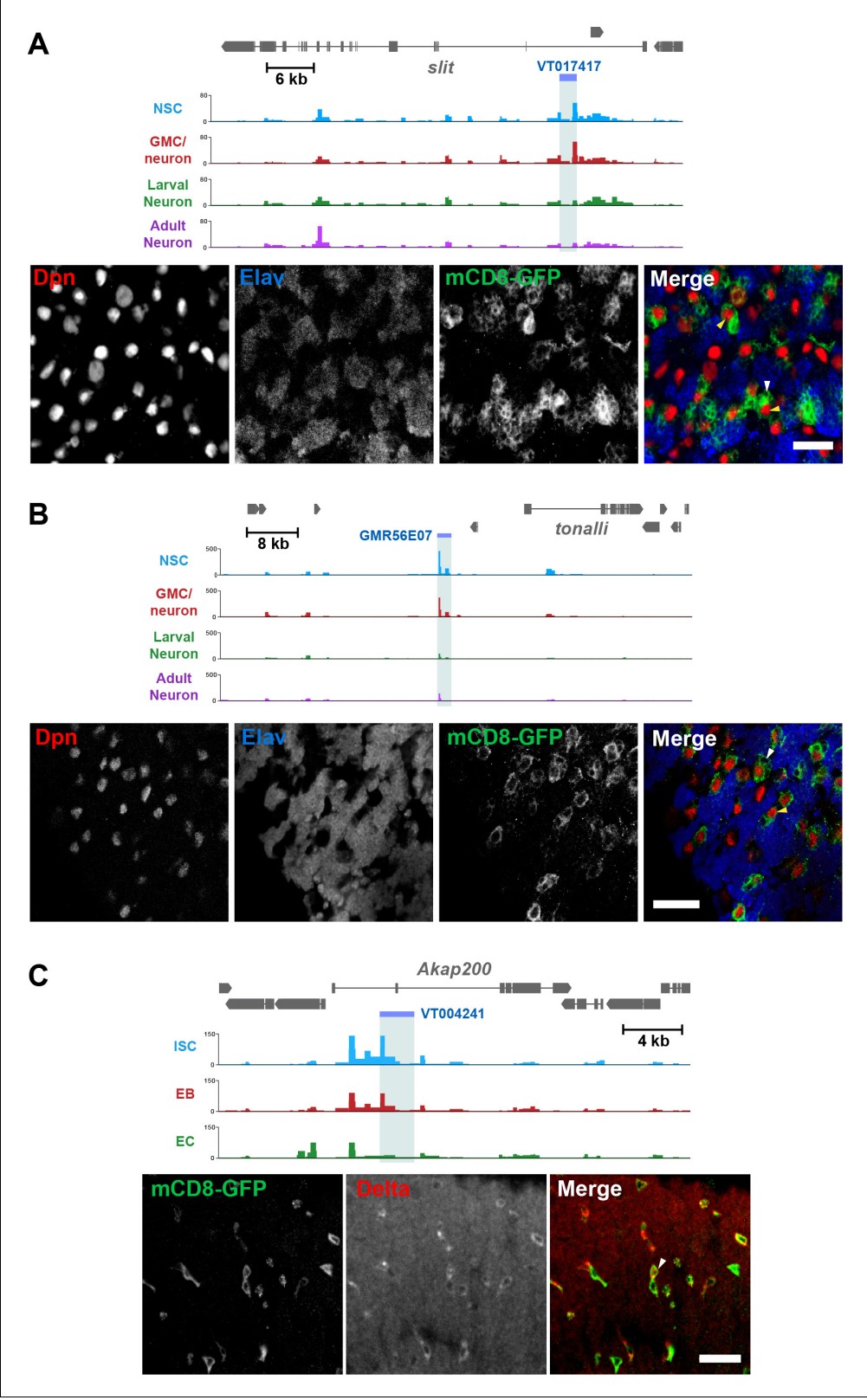

**Figure 6.** Identification of cell-type-specific enhancers from Dam accessibility data. (**A**) Intronic putative enhancer region VT017417 within *slit* locus reveals expression of GFP reporter gene predominantly in GMCs (white arrow) and newly born neurons, as well as some NSCs (Dpn positive, yellow arrow). (**B**) Intergenic putative enhancer region GMR56E07 shows expression of GFP reporter gene predominantly in NSCs (yellow arrow), with some GMC

*Figure 6 continued on next page*

*Figure 6 continued*

expression (white arrow). (**C**) Intronic putative enhancer region VT004241 shows expression of GFP reporter predominantly in the ISCs (marked with Delta, white arrow). All scale bars = 20 μm.

DOI: https://doi.org/10.7554/eLife.32341.019

(rpm) were relatively infrequent. In other words, most of the genome is inaccessible or accessible at very low levels, whilst hyper-accessible regions are comparatively rare in all cell types (**Figure 7A**). The greatest difference between the distributions of neurons and NSCs was apparent at very low read numbers. We observed that there was a significantly greater proportion of GATC fragments with low read counts but without being completely inaccessible (~1–3 rpm), in the NSC data compared to neurons (**Figure 7A**). This abundance of genomic regions in the stem cells with low-level

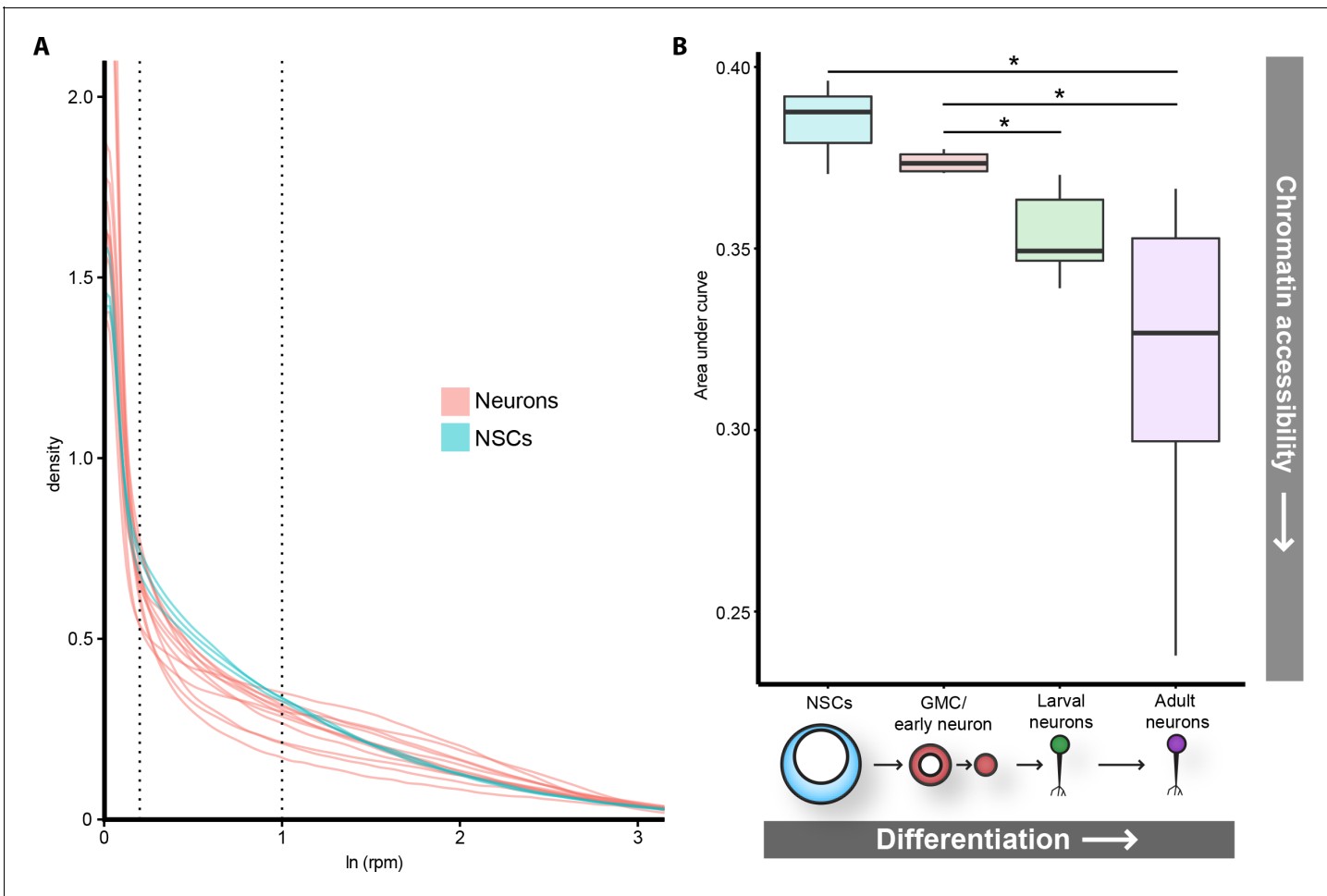

**Figure 7.** Global chromatin accessibility is reduced in differentiated neurons. (**A**) Log transformed distribution of read counts at GATC fragments for neuronal (pink) or NSC replicates (blue). In addition to adult neuron data described in previous figures, CATaDa data for cholinergic, glutamatergic, and GABAergic adult neurons are included. NSC data include extra replicate from (**Marshall et al., 2016**). (**B**) Areas under curve for region bound by dotted lines in (**A**). Corresponding to ~1–3 rpm. Data include extra neuronal and NSC replicates shown in (**A**), as well as corresponding replicates at L3 for gutamatergic, GABAergic, and cholinergic neurons. Note that area under curve corresponds to proportion of GATC fragments having mapped reads within indicated range (i.e. NSCs have ~38% (median) of all GATC fragments within 1–3 rpm mapped reads, compared to ~32% for adult neurons. Results were considered significant at *p<0.05.

DOI: https://doi.org/10.7554/eLife.32341.020

The following figure supplement is available for figure 7:

**Figure supplement 1.** Average distribution of sequencing reads for CNS and midgut cell types.
DOI: https://doi.org/10.7554/eLife.32341.021

chromatin accessibility indicates that open chromatin is more broadly distributed than in NSCs than neurons. We find that this trend is also apparent in the intermediate progenitor cell types, having intermediate amounts of GATC fragments mapping to low read counts (*Figure 7B*, *Figure 7—figure supplement 1A*). Conversely, we observed a trend towards greater number of GATC fragments to which zero reads were mapped as differentiation progresses (*Figure 7—figure supplement 1B*). These data demonstrate that differentiated cells are more likely to have regions of chromatin that are completely inaccessible to Dam, indicating a globally lower amount of accessible chromatin in neurons. These trends are also observed in the midgut cells, implying that global changes to accessible chromatin are a common feature of somatic stem cell lineages in vivo (*Figure 7—figure supplement 1C*).

## Discussion

Recent studies have provided insights into chromatin accessibility of individual cell-types using accessibility assays coupled with cell sorting (*Pearson et al., 2016*). Whilst these strategies have been proven to produce meaningful biological data, they suffer from being technically challenging, particularly with regards to cell isolation. We have demonstrated that CATaDa yields chromatin accessibility profiles for defined cell types in vivo without the need for cell isolation, fixation or the extraction of naked chromatin. In addition to its ease of use, CATaDa also has the advantage that the marking of accessible DNA occurs in vivo. Due to this, there are no artefacts associated with chemical fixation or washing of the chromatin prior to the assay (*Baranello et al., 2016*). Furthermore, widely used tissue dissociation protocols have recently been shown to result in substantial gene expression artefacts, a risk that is also circumvented by labelling the DNA in vivo (*van den Brink et al., 2017*). In addition, as Dam is expressed in vivo over several hours, the profiles produced will reflect dynamic changes to chromatin structure over the entire time period during which Dam is expressed.

CATaDa is limited by its resolution, which is restricted by the frequency of GATC sites in the genome (median spacing of ~200 bp in *Drosophila*). However, this can be increased by using a modified Dam in conjunction with immunoprecipitation (Dam-IP)(*Xiao et al., 2010*; *Xiao and Moore, 2011*). Due to the dependence of Dam for methylation of GATC sequences, biases may be observed at loci which are depleted for GATC. It is worth noting that extensive sequence biases have also been reported for DNAse and ATAC-seq (*Madrigal, 2015*). Although single cell protocols have recently been developed for chromatin accessibility techniques (*Jin et al., 2015a*; *Buenrostro et al., 2015*), for routine experiments it is more usual to require a relatively large number of cells. For example, for FAIRE-seq it is recommended to have a minimum of $1 \times 10^6$ cells (*Tsompana and Buck, 2014*; *Simon et al., 2013*), whilst DNase-seq typically requires $1 \times 10^7$ cells (*Tsompana and Buck, 2014*; *Song and Crawford, 2010*). In contrast, DamID experiments can be performed with as few as 1000 cells (*Tosti, 2017*), therefore CATaDa is likely to also be effective with low cell numbers, making it competitive with ATAC-seq (500–50,000 cells) (*Buenrostro et al., 2013*; *Marshall et al., 2016*). Furthermore, single cell DamID has also recently been demonstrated, indicating that the minimum number of cells required for CATaDa is one (*Kind et al., 2015*).

Although we see instances of false negatives in our peak identification with CATaDa due to lack of available GATC sites for methylation, we also observe a number of peaks unique to CATaDa. Whilst it is possible that these signals are the result of an experimental artefact, it seems likely that they are genuine accessible loci. This could be accounted for by the fact that Dam methylase is a relatively small protein compared to Tn5 transposase (utilised in ATAC-seq – 55 and 32 KDa respectively [*Naumann and Reznikoff, 2002*; *Boye et al., 1992*]), Therefore, Dam may be able to methylate sites which are inaccessible to Tn5. Alternatively, Dam may be able to methylate some nucleosome bound DNA in which the GATC site is exposed to the nucleoplasm, which may be insufficient for transposition. Whilst the presence of some potential false positives and negatives in the data may be a problem for some applications, overall, CATaDa produces similar overall results to ATAC and FAIRE-seq. As there is already a significant amount of disagreement between ATAC and FAIRE-seq, it is impossible to say whether a unique peak called by any method is a true peak or not, and as these techniques work via different principles, it is likely that these differences reflect biases unique to each approach.

Due to the technical differences between CATaDa and currently favoured alternatives (e.g. ATAC-seq), it is clear that the choice of which of these techniques is most appropriate is dependent on the application in question. The limited resolution of CATaDa means that it is not well suited for identifying precise limits of enhancer regions or nucleosome positions when compared to ATAC-seq. However, we have demonstrated that CATaDa resolution is sufficient to investigate broad differences in chromatin accessibility between samples, to identify enriched sequence motifs, and even to identify individual cell-type-specific enhancers. Therefore, the technique is suited to answering questions regarding the biology of cell-types of interest. The requirement for expression of Dam may represent difficulties for non-genetically-tractable model systems, or complicate experimental design if a mutant background is desired, (although it should be noted that in the latter case, a genetically encoded cell-type-specific marker may have to be included anyway to facilitate cell-separation for ATAC-seq or other approaches – therefore a complicated genetic background may be unavoidable). On the other hand, the advantages of being able to assay individual cell types residing within a complex tissue, are clear. If the driver used is specific enough, this may even be achieved with minimal or no dissection.

Targeted DamID is rapidly being embraced by the *Drosophila* community with, at present, over 135 laboratories having requested the reagents and a number of papers already published (*Southall et al., 2013*; *Dinges et al., 2017*; *Marshall and Brand, 2017*; *Cheetham and Brand, 2018*; *Jin et al., 2015b*; *Spéder and Brand, 2018*). Progress is also being made in adapting it for use in vertebrate models (*Tosti, 2017*). Whenever the binding of a protein of interest (POI) is investigated with this technique, Dam-only data (representing chromatin accessibility) for the cell type being assayed is also generated, as it is the control for which the Dam-POI is normalised. Therefore, researchers performing DamID experiments can now take advantage of this data, getting a '2-for-1 deal' whenever they use TaDa to profile the binding of a POI. Furthermore, much of these data are already available from published studies that could be readily analysed to provide novel biological insights.

The sequence of events which lead to repression of open chromatin in the transition between stem cells and their progeny is not well defined. Through identification of functional elements of the genome at various stages in this process, we can begin to understand how the dynamic chromatin landscape impacts the regulation and maintenance of differentiated cell states. Interestingly, we observe that chromatin accessibility in intermediate cell types is broadly more similar to their stem cell precursors than their differentiated progeny in CNS and midgut lineages (*Figure 4B,C*, *Figure 5B,C,F*). This indicates that many stem-cell-specific regulatory regions remain accessible in intermediate cell types and are not fully repressed until terminal differentiation. This seems particularly surprising in the case of the EBs in the midgut considering that these cells do not undergo further mitotic divisions and are committed to a particular cell fate before their genesis by Notch signalling in the ISC (*Ohlstein and Spradling, 2007*). Furthermore, in the intermediate cells of the larval CNS (GMCs and immature neurons), a relatively high proportion of cells profiled are neurons, which express markers thought to be associated with fully differentiated cells, suggesting that these regulatory regions may be open prior to terminal fate specification.

The reason for these regions of chromatin remaining accessible in stem cell progeny is unclear, however there are several plausible explanations. Firstly, regions that are bound by transcription factors that activate transcription may be replaced by transcriptional repressors. Such repressive factors are known to have detectable open chromatin 'footprints', similarly to activating factors (*Mall et al., 2017*; *Dinges et al., 2017*). Alternatively, the same factors that bind in the stem cells may recruit new binding partners that alter their activity to initiate repression rather than activation of gene expression. This explanation would require that further modifications occur to the chromatin following repressor activity as open chromatin regions are lost in fully differentiated cells, suggesting that repressors may no longer be bound. Retention of open chromatin regions in intermediate cell types may also reflect increased plasticity, indicating that cell fate has not yet been fully determined and that lineage reversion or dedifferentiation is possible given the introduction of the correct combination of factors. This idea is supported by the fact that immature post-mitotic neurons have been experimentally induced to dedifferentiate by interventions that are ineffective in fully differentiated adult cells (*Southall et al., 2014*; *Marshall and Brand, 2017*). It has been suggested that in some physiological contexts differentiated cells may revert to replenish stem cell pools (*Kai and*

*Spradling, 2004*; *Yan et al., 2017*; *Cheetham and Brand, 2018*; *Jin et al., 2015b*). This idea may help to explain retention of plasticity in these cell types.

It is widely asserted that terminally differentiated cells have limited accessible chromatin whilst their progenitors maintain a broadly open chromatin landscape. However, there are few studies that investigate this phenomenon in vivo. Furthermore, the chromatin state in lineage committed intermediate progenitor cell types has been little studied. With CATaDa we have acquired evidence to indicate that NSCs in the developing *Drosophila* CNS appear to have more broadly open chromatin landscapes than their fully differentiated progeny. These differences are predominantly in the range of 1 to 3 rpm, which reflects chromatin with very low accessibility. Furthermore, the intermediate or immature progenitors, in both the CNS and midgut, retain a relatively open chromatin state, similar to that of their stem cell precursors (*Figure 4B–C*, *Figure 5B,D,F*). Together, these data support the model of stem cells containing more chromatin that has the potential to be accessed, whereas in differentiated cells, this flexibility is reduced and more regions are rendered completely inaccessible. In pluripotent stem cells, the more highly accessible chromatin landscape is thought to promote rapid initiation of multiple gene expression programmes. The fact that this is also a feature of somatic stem cells suggests that these cells may retain an unexpected level of plasticity even after their terminal cell fate has been specified.

In conclusion, we have shown that cell-type-specific chromatin accessibility profiles can be obtained through tightly controlled expression of Dam methylase. These data can be used to predict cell-type-specific enhancers, as well as gaining insights into the global regulation of chromatin. Of particular interest are the dynamic changes in accessibility as cells progress towards terminal differentiation (e.g. the unique open regions observed in GMCs/early neurons) and the delayed compaction of stem cell gene associated chromatin. Also, analysis of the genome-wide distribution of chromatin accessibility supports a model of gradual compaction of large regions of low accessibility chromatin during differentiation. Overall, our results from profiling developing cell types illuminate the dynamic nature of chromatin accessibility in differentiation, and hint at organising principles which may apply to all somatic stem-cell lineages.

## Materials and methods

### Fly stocks

*tub-GAL80^{ts}; UAS-LT3-NDam* (*Southall et al., 2013*) was used to allow cell-specific expression of *Dam*. The following *GAL4* driver lines were used to drive *Dam* expression in the CNS: *wor-GAL4* (*Albertson et al., 2004*) for neuroblasts, *GMR71C09-GAL4* (Bloomington #39575) for GMCs and newly born neurons and *nSyb-GAL4* (Bloomington #51941) for mature larval and adult neurons. For expression of *Dam* in the gut the following lines were used for ISC, EB, and EC expression, respectively: *esg-GAL4, UAS-2xEYFP/Cyo; Su(H)GBE-GAL80/TM3 Sb, Su(H) GBE-GAL4, UAS-CD8GFP/Cyo* and *{GawB}Myo31DF^{NP0001}/CyO* (*Wang et al., 2014*; *Jiang et al., 2009*). *P{tubP-GAL4}LL7/TM6, Tb* (Bloomington #5138) was used to drive ubiquitous expression in antennal-eye discs.

*Cha^{MI04508}-T2A-GAL4, Gad1^{MI09277}-T2A-GAL4* and *vGlut^{MI04979}-T2A-GAL4* driver lines were used to drive expression in cholinergic, GABAergic and glutamatergic neurons, respectively (*Diao et al., 2015*).

### Chromatin accessibility profiling using targeted DamID (CATaDa)

To induce tissue-specific *Dam* expression, *GAL4* driver lines were crossed to *GAL80^{ts}; UAS-LT3-NDam* virgin females. Embryos were collected for 4 hr then raised at 18°C. Animals were transferred to 29°C at either 7 days after embryo deposition for 24 hr to obtain third instar larval tissues, or three days after eclosion to obtain adult heads. Fifty brains or thirty antennal-eye discs were dissected in PBS with 100 mM EDTA for each replicate. *71C09-GAL4 > UAS-LT3-NDam* ventral nerve cords (VNCs) were dissected and central brain and optic lobe regions discarded due to presence of observed *71*C09-GAL4 expression in a small subset of central brain neuroblasts. For midgut experiments, Animals were transferred to 29°C for 24 hr at three days after eclosion, and thirty midguts dissected per genotype. The gut regions dissected were between the crop and malpigian tubules. The amount of tissue dissected was chosen to ensure the presence of an appropriate number of Dam-expressing cells (~10000 (*Southall et al., 2013*)). (For this study, we chose to dissect all tissues

due to non-specificity of some drivers, however, this should not be required if driver expression is specific). Genomic DNA extraction and sequencing library preparation was performed as described previously (*Marshall et al., 2016*), with minor modifications - MyTaq (Bioline) was used for PCR amplification of adapter ligated DNA. Libraries were sequenced using Illumina HiSeq single-end 50 bp sequencing. Two replicates of at least 10 million reads were acquired for each cell type. A third replicate was acquired for *GMR71C09-GAL4* to allow for comparisons in *Figure 7*. Sequencing data were mapped back to release 6.03 of the Drosophila genome using a previously described pipeline, which was modified to output Dam-only datafiles (*Marshall and Brand, 2015*) (available at https://github.com/tonysouthall/damidseq_pipeline_output_Dam-only_data [*Southall, 2017a*; copy archived at https://github.com/elifesciences-publications/damidseq_pipeline_output_Dam-only_data]). This includes mapping reads to the genome using bowtie and assigning to bins delimited by GATC sites.

## Immunohistochemistry and imaging

*71C09-GAL4 > UAS-mCD8-GFP* third instar larval CNS or adult midgut were dissected in PBS and fixed for 20 min with 4% formaldehyde in PBS, 0.5 mM EGTA, 5 mM $MgCl_2$. Tissues were stained with rat anti-elav (Developmental Studies Hybridoma Bank), chicken anti-GFP (Thermo scientific), mouse anti-Delta (Developmental Studies Hybridoma Bank), and guinea pig anti-deadpan (kind gift from A. Brand). Samples were imaged using a Zeiss LSM510 confocal microscope.

## Peak calling

Peaks were called and mapped to genes using a custom Perl program (available at https://github.com/tonysouthall/Peak_Calling_for_CATaDa [*Southall, 2017b*; copy archived at https://github.com/elifesciences-publications/Peak_Calling_for_CATaDa]). In brief, a false discovery rate (FDR) was calculated for peaks (formed of two or more consecutive GATC fragments) for the individual replicates. Then, each potential peak in the data was assigned a FDR. Any peaks with less than a 1% FDR were classified as significant. Significant peaks present in both replicates were used to form a final peak file. Any gene (genome release 6.11) within 5 kb of a peak (with no other genes in between) was identified as a potentially regulated gene.

## Motif identification

Regions of differentially accessible chromatin were identified by calculating GATC fragments for which a difference of >20 rpm was observed between every replicate between at least two cell types. ~97% of these fragments lie within statistically significant (FDR < 1%) different (between respective cell types) regions. These regions were identified by running the peak calling program on comparison gff files (generated by subtraction with negative values zeroed). The other ~3% are GATC fragments that show >20 rpm difference in isolation, therefore are not identified by the peak calling program. Hierarchical clustering was performed using Morpheus (*Broad Institute, 2017*). Sequences for major clusters showing enrichment for a given cell type were then analysed using MEME-ChIP (*Bailey et al., 2009*).

## Gene ontology analysis

Potentially regulated genes, with a peak height of at least 10 rpm were submitted to GOToolBox (*Martin et al., 2004*) for GO analysis. GO term enrichments (frequency in data set divided by expected frequency) were calculated for each cell type.

## Cell-type-specific enhancer prediction

GATC fragments were identified with at least a 10 rpm difference in all replicates between at least two cell types. Any peak >2 kb from a transcriptional start site that did not overlap with coding sequence was designated an enhancer. Enhancers satisfying these criteria, which were covered by an available Vienna VT (*Kvon et al., 2014*) or Janelia FlyLight (*Jenett et al., 2012*) GAL4 line were chosen for validation based on the magnitude of the change in accessibility.

## Statistical analysis and data presentation

For comparing Dam data with ATAC and FAIRE, Monte Carlo experiments were performed using a custom Perl script (available at https://github.com/tonysouthall/Monte_Carlo_simulation [*Southall, 2017c*; copy archived https://github.com/elifesciences-publications/Monte_Carlo_simulation]). Comparisons of areas under curve in *Figure 7* were performed using Welch's ANOVA for heteroscedasticity with Games-Howell post-hoc test in R. For data with equal variances, ANOVA was used with Tukey post-hoc testing (e.g. *Figure 7—figure supplement 1*). Results were considered significant at *$p < 0.05$, **$p < 0.01$. Principal component analyses and correlation matrix plots were produced using deepTools (*Ramírez et al., 2014*). Average TSS signal profiles were made using SeqPlots R/Bioconductor package (*Stempor and Ahringer, 2016*). All other figures were produced using the ggplot2 package in R.

## Acknowledgements

We would like to thank Seth Cheetham, Jelle van den Ameele, and members of the Southall group for feedback and advice on this project. We are also grateful to Owen Marshall for his helpful discussions during the preparation of the manuscript and analysis of the data. We would like to thank Julia Cordero, Matthias Landgraf, Holly Ironfield and Benjamin White for providing fly stocks. Other stocks obtained from the Bloomington Drosophila Stock Center (NIH P40OD018537) and the Vienna Drosophila Resource Center (VDRC, www.vdrc.at). We also thank Andrea Brand for the Deadpan antibody. This work was funded by Wellcome Trust Investigator grant 104567 to TDS.

## Additional information

### Funding

| Funder | Grant reference number | Author |
| --- | --- | --- |
| Wellcome Trust | Wellcome Trust Investigator Grant 104567 | Tony D Southall |

The funders had no role in study design, data collection and interpretation, or the decision to submit the work for publication.

### Author contributions

Gabriel N Aughey, Conceptualization, Data curation, Formal analysis, Validation, Investigation, Visualization, Methodology, Writing—original draft, Project administration, Writing—review and editing; Alicia Estacio Gomez, Investigation, Writing—review and editing; Jamie Thomson, Hang Yin, Validation, Investigation; Tony D Southall, Conceptualization, Data curation, Formal analysis, Supervision, Funding acquisition, Investigation, Methodology, Writing—original draft, Project administration, Writing—review and editing

### Author ORCIDs

Gabriel N Aughey http://orcid.org/0000-0001-5610-9345
Tony D Southall http://orcid.org/0000-0002-8645-4198

### Decision letter and Author response

Decision letter https://doi.org/10.7554/eLife.32341.028
Author response https://doi.org/10.7554/eLife.32341.029

## Additional files

### Supplementary files

• Transparent reporting form
DOI: https://doi.org/10.7554/eLife.32341.022

## Major datasets

The following dataset was generated:

| Author(s) | Year | Dataset title | Dataset URL | Database, license, and accessibility information |
|---|---|---|---|---|
| Aughey GN, Estacio Gomez A, Southall TD | 2018 | CATaDa chromatin accessiblity data for neural and midgut cell types. | https://www.ncbi.nlm.nih.gov/geo/query/acc.cgi?acc=GSE104801 | Publicly available at the NCBI Gene Expression Omnibus (accession no. GSE104801) |

The following previously published dataset was used:

| Author(s) | Year | Dataset title | Dataset URL | Database, license, and accessibility information |
|---|---|---|---|---|
| Marshall OJ, Brand AH | 2015 | damidseq_pipeline: an automated pipeline for processing DamID sequencing datasets | https://www.ncbi.nlm.nih.gov/geo/query/acc.cgi?acc=GSE69184 | Publicly available at the NCBI Gene Expression Omnibus (accession no. GSE69184) |

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
