## [Decision Letter]

Thank you for submitting your article "CATaDa reveals global remodelling of chromatin accessibility during stem cell differentiation in vivo" for consideration by *eLife*. Your article has been favorably evaluated by K VijayRaghavan (Senior Editor) and three reviewers, one of whom is a member of our Board of Reviewing Editors. The following individual involved in review of your submission has agreed to reveal their identity: Gabriel E. Zentner (Reviewer #3).

The reviewers have discussed the reviews with one another and the Reviewing Editor has drafted this decision to help you prepare a revised submission. From the reviews, and discussion between the reviews, there was a consensus that the paper has greatest value in the category of "Tools and Resources", which is a specific type of short-format *eLife* paper. Thus I'd like to propose that we publish your paper, after revisions as discussed here and in the reviews, as a "Tools and Resources" article. Please consider that publishing in this category will in no way diminish the import of your work. On the contrary, since CATaDa is a unique and generally applicable tool, we expect it to be highly appreciated by our readership and widely cited. As to revisions, one thing all the reviewers agreed upon is the paper should include a more thorough, genome-wide comparison of the CATaDa, ATAC, and FAIRE methods for chromatin profiling, and discussion of the pros and cons of CATaDa in relative terms. This is important as ATAC and FAIRE are already widely used. A number of other suggested revisions can be found in the reviews, below. We hope you will consider publishing this interesting work as a Tools and Resources report.

*Reviewer #1:*

The technique detailed in the manuscript is an interesting alternative to currently used methods for chromatin analysis such as ATAC- and FAIRE-seq. In particular the possibility to forego the requirement to isolate the cell type of interest is very attractive, as it would speed up the process, thus limiting the time required for tissue manipulation prior to chromatin extraction. However, we feel that the following points could be improved:

• A more complete, genome-wide comparison between CATaDa and ATACseq and FAIREseq is necessary to make this a useful into to this new method.

• As different driver lines are used, it would be beneficial to test the expression levels of the Dam in the various cell types here discussed.

• The technique is stated to be quite reproducible and Figure 2—figure supplement 1 is introduced to support this. However, it is not explained what samples were plotted and also whether this correlation has been similarly observed for all samples and replicates or the samples plotted are simply the two showing the best correlation observed. Also, a similar plot for midgut cells would be required.

• In the third paragraph of the subsection “CATaDa profiling shows dynamic changes in chromatin accessibility during differentiation of the nervous system”, it is shown that "GMC/ newly born neuron profiles frequently show intermediate signal" at several loci where gene expression is activated/inactivated during development. In how many of the loci where such behavior is experimentally known to occur can this be observed? Can similar patterns be identified with ATAC- and FAIRE-seq just as well?

• In the fifth paragraph of the subsection “CATaDa profiling shows dynamic changes in chromatin accessibility during differentiation of the nervous system”, is discussed the ability to identify sequence motifs in regions of the NSC genome classified as open. However, only one example of identified motif (i.e. E-box) is reported. What other known motifs have been identified? Also, in regards of the motifs for which no known transcription factor could be identified, how likely are they to be true binding motifs rather than random sequences?

• In regards to the Gene Ontology analysis described in the last paragraph of the subsection “CATaDa profiling shows dynamic changes in chromatin accessibility during differentiation of the nervous system”, it would be good if a supplementary table could be added, reporting the top categories and their relative significance of enrichment, rather than just list a few terms that were expected.

• The comparison described in the last paragraph of the subsection “Chromatin accessibility in adult midgut cell types”, between brain and midgut cells feels superfluous. Although it's good that brain and midgut cells cluster in two different groups, it is fairly obvious given the very different function of the two organs. This paragraph could be rewritten by giving more space to the direct comparison of NSC and ISC, highlighting similarities in the chromatin in the vicinity of genes (e.g.) related to growth and proliferation.

• Subsection “Enhancer prediction from Dam Accessibility data”, last paragraph, it would be helpful to know how many reporter lines were tested and how many matched the prediction of enhancer activity. Also, this would be a good opportunity for further comparison with ATAC- and FAIRE-seq.

• In Figure 5, what cell type are the GFP+ Dl- cells?

• Subsection “Distribution of sequencing reads reveals changes in global chromatin accessibility during lineage development”, second paragraph, it is stated that Figure 7—figure supplement 1 was analyzed via "visual inspection". Some form of statistical test would go a long way to improve the reported observation.

• When comparing GATC fragments as described in the "Materials and methods" section ("Motif identification" and "Cell type specific enhancer prediction" sections), it is not stated whether the difference in RPM values were selected via statistical tests or nor. In the latter case, it would be best to use the replicates in a statistical test to evaluate "real" differences between cell types.

*Reviewer #2:*

Various chromatin accessibility methods have been used to map potential regulatory elements, beginning with DNase I hypersensitive sites to more recent FARE and ATAC-seq methods. These methods can be useful when combined with other strategies that more directly report on the function of regulatory elements. In this study, the authors develop a method (CATaDa) to map open chromatin regions based on tissue-specific expression of the *E. coli* Dam methylase in *Drosophila* followed by mapping of GATC methylation sites. The main advantage of the method over previous approaches is that it requires fewer cells and does not require isolation of cells of interest. Using this method the authors profile chromatin accessibility of specific cell types in the *Drosophila* neuronal and midgut lineages and find changes in chromatin accessibility at different developmental stages.

The method the authors have developed clearly has some advantages over ATAC-seq, probably the most popular current method for assays of genome-wide chromatin accessibility. However, unlike ATAC-seq, this method requires the generation of transgenic animals that express Dam in specific cell types. This may limit its general use but the authors report great interest from the *Drosophila* community (based on requests for reagents). As acknowledged by the authors, the resolution of the method is limited by the occurrence of GATC sites (1/~250 bp). My main concern is that the profiling the authors report provides limited functional insight into the roles of specific enhancers or other regulatory elements in regulation of cell type-specific gene expression. The significance of the global changes in chromatin accessibility described by the authors also remains unclear.

*Reviewer #3:*

Chromatin accessibility is a hallmark of active regulatory regions and is dynamic during development. While many chromatin profiling approaches have been implemented in cultured cells, mapping chromatin accessibility in tissues has been more difficult due to the need to physically dissociate target cells from the organism of interest. Here, the authors present CATaDa, a variation on their earlier TaDa technique for tissue-specific in vivo profiling of protein-DNA interactions. CATaDa makes use of untethered Dam's propensity for methylating open regions of chromatin as a means to assay chromatin accessibility. The authors test CATaDa in neural and midgut stem cells from *Drosophila*, finding that chromatin accessibility changes during differentiation and is ultimately reduced in terminally differentiated cells. CATaDa also facilitated the detection of cell type-specific enhancer elements. I think the paper is a nice demonstration that Dam works as a reagent for tissue-specific profiling of chromatin accessibility in vivo (the stepwise changes in GO terms enriched in NSCs to neurons are particularly nice). The fact that CATaDa does not require cell isolation prior to chromatin profiling as FAIRE or ATAC-seq does makes it a strong choice for this purpose. I think the method will be useful, though I would like to see more extensive comparison to FAIRE and ATAC-seq.

My read of the manuscript is that CATaDa was a hammer in search of a nail, which turned out to be stem cell differentiation. My feeling is that the biological aspects of the story are case studies for the method (though it is of course good to have confirmation that chromatin accessibility generally decreases during stem cell differentiation in vivo) rather than a story in their own right. I am therefore inclined to think that the paper would be better suited to being a Tools and Resources article, as both the method and the data generated would be useful for the community.

---

## [Author Response]

Reviewer #1:The technique detailed in the manuscript is an interesting alternative to currently used methods for chromatin analysis such as ATAC- and FAIRE-seq. In particular the possibility to forego the requirement to isolate the cell type of interest is very attractive, as it would speed up the process, thus limiting the time required for tissue manipulation prior to chromatin extraction. However, we feel that the following points could be improved:• A more complete, genome-wide comparison between CATaDa and ATACseq and FAIREseq is necessary to make this a useful into to this new method.

We recognise the need for further comparisons between these methods. To address this, we have included the following new analyses in our revised manuscript.

- New analysis of genome wide comparison of intersecting CATaDa peaks with ATAC and FAIRE – including highlighting differences between peaks at promoters vs. other genomic loci (Figure 2).

- Demonstrated increased frequency of CATaDa signal in proximity to ATAC/FAIRE peaks (Figure 2).

- We investigate the prevalence of peaks appearing only in CATaDa data using different stringencies for peak calling (Figure 2—figure supplement 2).

- We account for much of the difference between methods by analysing GATC site abundance at different genomic loci (Figure 2—figure supplement 3).

- We show correlation between ATAC and CATaDa peaks (Figure 2—figure supplement 2).

- Further discussion of differences between techniques.

• As different driver lines are used, it would be beneficial to test the expression levels of the Dam in the various cell types here discussed.

The ribosome re-initiation technique results in levels of Dam that are so low that it is undetectable by any of the means at our disposal (see Southall et al., 2013). Furthermore, we have no data to suggest that the rate of Dam translation relates to transgene expression in a linear manner. Therefore, even if we were able to compare the expression levels of the GAL4 drivers between individual cell types in the tissues in question, this would not necessarily be informative in regards to the levels of Dam protein present in these cell types. If high expression of GAL4 drivers were causing Dam to be expressed at a level that is too high, this would be apparent as we would not see the dynamic range of methylation signal observed in all cell types, as Dam methylation would be saturating.

• The technique is stated to be quite reproducible and Figure 2—figure supplement 1 is introduced to support this. However, it is not explained what samples were plotted and also whether this correlation has been similarly observed for all samples and replicates or the samples plotted are simply the two showing the best correlation observed. Also, a similar plot for midgut cells would be required.

We have updated and expanded Figure 2—figure supplement 1 to include details of replicate correlations for all groups studied.

• In the third paragraph of the subsection “CATaDa profiling shows dynamic changes in chromatin accessibility during differentiation of the nervous system”, it is shown that "GMC/ newly born neuron profiles frequently show intermediate signal" at several loci where gene expression is activated/inactivated during development. In how many of the loci where such behavior is experimentally known to occur can this be observed? Can similar patterns be identified with ATAC- and FAIRE-seq just as well?

To our knowledge, there are no available RNA-seq datasets for these cell types (i.e. GMCs), which would be necessary for us to be able to define such loci. Therefore it is difficult to make these kinds of conclusions on a larger scale than the examples which we have highlighted. Likewise, there are no chromatin accessibility data available (again to our knowledge), therefore it is not possible to make this comparison. For us to perform these experiments would be beyond the remit of this study.

• In the fifth paragraph of the subsection “CATaDa profiling shows dynamic changes in chromatin accessibility during differentiation of the nervous system”, is discussed the ability to identify sequence motifs in regions of the NSC genome classified as open. However, only one example of identified motif (i.e. E-box) is reported. What other known motifs have been identified? Also, in regards of the motifs for which no known transcription factor could be identified, how likely are they to be true binding motifs rather than random sequences?

We have included a new supplementary figure ((Figure 4—figure supplement 3) showing the top motifs identified for each group. The e-values reported in the new figure reflect the chance of any given motif occurring randomly. Given that many of these motifs occur at a rate much higher than is expected by chance, it is reasonable to assume that they are functional in the genome, and are therefore likely to be recognised by extrinsic factors.

• In regards to the Gene Ontology analysis described in the last paragraph of the subsection “CATaDa profiling shows dynamic changes in chromatin accessibility during differentiation of the nervous system”, it would be good if a supplementary table could be added, reporting the top categories and their relative significance of enrichment, rather than just list a few terms that were expected.

We have added a supplementary table (Figure 4—figure supplement 5) with these data as requested.

• The comparison described in the last paragraph of the subsection “Chromatin accessibility in adult midgut cell types”, between brain and midgut cells feels superfluous. Although it's good that brain and midgut cells cluster in two different groups, it is fairly obvious given the very different function of the two organs. This paragraph could be rewritten by giving more space to the direct comparison of NSC and ISC, highlighting similarities in the chromatin in the vicinity of genes (e.g.) related to growth and proliferation.

We have rephrased parts of this paragraph as suggested, and added a new supplementary figure (Figure 5—figure supplement 2) highlighting the similarity in the vicinity of loci involved in growth and proliferation as suggested.

• Subsection “Enhancer prediction from Dam Accessibility data”, last paragraph, it would be helpful to know how many reporter lines were tested and how many matched the prediction of enhancer activity. Also, this would be a good opportunity for further comparison with ATAC- and FAIRE-seq.

We have amended the manuscript to address this point.

• In Figure 5, what cell type are the GFP+ Dl- cells?

Based on the size and morphology of these cells they are probably enteroblasts (EBs). This is supported by the fact that they seem to appear in proximity to the ISCs. The figure legend has been updated to reflect this.

• Subsection “Distribution of sequencing reads reveals changes in global chromatin accessibility during lineage development”, second paragraph, it is stated that Figure 7—figure supplement 1 was analyzed via "visual inspection". Some form of statistical test would go a long way to improve the reported observation.

We believe that the way this sentence was written in our previous submission may have been misleading. We initially observed this trend with a limited number of datasets, which we included as a supplementary figure, however we were unable to provide a meaningful statistical comparison until we added more replicates to the analysis as explained in the subsequent paragraph. The statistical in question that supports our observation is included in Figure 7.

We have re-written this part of the manuscript for improved clarity.

• When comparing GATC fragments as described in the "Materials and methods" section ("Motif identification" and "Cell type specific enhancer prediction" sections), it is not stated whether the difference in RPM values were selected via statistical tests or nor. In the latter case, it would be best to use the replicates in a statistical test to evaluate "real" differences between cell types.

~97% of the fragments with > 20 RPM difference lie within statistically significant (FDR < 1%) different regions. The other ~3% are GATC fragments that show >20 rpm difference in isolation, therefore are not identified by our peak calling program and do not have an FDR assigned. We have clarified this point in the Materials and methods section.

Reviewer #2:[…] The method the authors have developed clearly has some advantages over ATAC-seq, probably the most popular current method for assays of genome-wide chromatin accessibility. However, unlike ATAC-seq, this method requires the generation of transgenic animals that express Dam in specific cell types. This may limit its general use but the authors report great interest from the Drosophila community (based on requests for reagents). As acknowledged by the authors, the resolution of the method is limited by the occurrence of GATC sites (1/~250 bp). My main concern is that the profiling the authors report provides limited functional insight into the roles of specific enhancers or other regulatory elements in regulation of cell type-specific gene expression. The significance of the global changes in chromatin accessibility described by the authors also remains unclear.

We have demonstrated that cell specific chromatin accessibility profiling with CATaDa allows the identification of cell specific enhancers (see Figure 6), despite the method having a lower resolution compared to other methods. We have also shown that enriched motifs can be identified in regions of the genome that are specifically accessible in individual cell types (see (Figure 4 and Figure 4—figure supplement 3 and Figure 4—figure supplement 4)). Further functional insight into individual enhancers would require cloning, dissection and/or mutation, which would also be required if using other accessibility profiling methods.

Regarding the significance of the global changes in chromatin accessibility, we have included some further discussion in our revised manuscript.